# Transcriptional Regulation of Yin-Yang 1 Expression through the Hypoxia Inducible Factor-1 in Pediatric Acute Lymphoblastic Leukemia

**DOI:** 10.3390/ijms23031728

**Published:** 2022-02-02

**Authors:** Gabriela Antonio-Andres, Gustavo U. Martinez-Ruiz, Mario Morales-Martinez, Elva Jiménez-Hernandez, Estefany Martinez-Torres, Tania V. Lopez-Perez, Laura A. Estrada-Abreo, Genaro Patino-Lopez, Sergio Juarez-Mendez, Víctor M. Davila-Borja, Sara Huerta-Yepez

**Affiliations:** 1Unidad de Investigación en Enfermedades Oncológicas, Hospital Infantil de México, Federico Gómez, Mexico City 06720, Mexico; gabya_24@yahoo.com.mx (G.A.-A.); ixnergal@gmail.com (M.M.-M.); stephy.mt12@gmail.com (E.M.-T.); valecita_vale@hotmail.com (T.V.L.-P.); 2División de Investigación, Facultad de Medicina, Universidad Nacional Autónoma de México, Mexico City 04510, Mexico; gustavobambam@gmail.com; 3Servicio de Hemato-Oncología, Hospital Infantil de Moctezuma, Mexico City 15530, Mexico; elvajimenez@yahoo.com; 4Consejo Nacional de Ciencia y Tecnología (CONACYT), Mexico City 03940, Mexico; 5Laboratorio de Investigación en Inmunología y Proteómica, Hospital Infantil de México, Federico Gómez, Mexico City 06720, Mexico; lauraestrada13@yahoo.com.mx (L.A.E.-A.); gena23pat@yahoo.com (G.P.-L.); 6Laboratorio de Oncología Experimental, Instituto Nacional de Pediatría, S.S.A., Mexico City 04530, Mexico; sjuarezm@pediatria.gob.mx (S.J.-M.); latrans86@hotmail.com (V.M.D.-B.)

**Keywords:** HIF-1α, YY1, ALL

## Abstract

Yin-Yang transcription factor 1 (YY1) is involved in tumor progression, metastasis and has been shown to be elevated in different cancers, including leukemia. The regulatory mechanism underlying YY1 expression in leukemia is still not understood. Bioinformatics analysis reveal three Hypoxia-inducible factor 1-alpha (HIF-1α) putative binding sites in the YY1 promoter region. The regulation of YY1 by HIF-1α in leukemia was analyzed. Mutation of the putative YY1 binding sites in a reporter system containing the HIF-1α promoter region and CHIP analysis confirmed that these sites are important for YY1 regulation. Leukemia cell lines showed that both proteins HIF-1α and YY1 are co-expressed under hypoxia. In addition, the expression of mRNA of YY1 was increased after 3 h of hypoxia conditions and affect several target genes expression. In contrast, chemical inhibition of HIF-1α induces downregulation of YY1 and sensitizes cells to chemotherapeutic drugs. The clinical implications of HIF-1α in the regulation of YY1 were investigated by evaluation of expression of HIF-1α and YY1 in 108 peripheral blood samples and by RT-PCR in 46 bone marrow samples of patients with pediatric acute lymphoblastic leukemia (ALL). We found that the expression of HIF-1α positively correlates with YY1 expression in those patients. This is consistent with bioinformatic analyses of several databases. Our findings demonstrate for the first time that YY1 can be transcriptionally regulated by HIF-1α, and a correlation between HIF-1α expression and YY1 was found in ALL clinical samples. Hence, HIF-1α and YY1 may be possible therapeutic target and/or biomarkers of ALL.

## 1. Introduction

Acute lymphoblastic leukemia (ALL) is the most common childhood malignancy worldwide, with an incidence of around 4.8 cases per 100,000 population aged 0 to 19 years. Around 2500 cases of ALL are diagnosed annually in the USA [1,2,3,4]. Advancements in treatment have been remarkable and have seen a 70–85% increase in survival rate [5,6]. Standard-risk patients have a 4-year event-free survival of 70–80%. However, the potential development of chemoresistance still represents a main obstacle for ALL treatment [7,8].

One of the mechanisms involved in the chemoresistance of tumors is the HIF-1. HIF-1 plays a very important role in cancer biology, participating in processes such as angiogenesis, maintenance of stem cells, metabolic reprogramming, epithelial-mesenchymal transition, as well as invasion, metastasis and resistance to radiation therapy and chemotherapy. HIF-1 levels correlate with tumor growth, vascularization and metastasis in both animal models and clinical studies [9,10,11,12,13].

HIF-1 is a heterodimeric protein complex composed of two subunits; one constitutively stable and expressed HIF1β, and the other inducible by O_2_ and growth factors, HIF-1α. HIF-1α is post-translationally modified by prolyl-hydroxylases within the oxygen dependent degradation domains (ODD), which promote the binding of pVHL (von Hippel-Lindau protein) and subsequent degradation via the proteasome [14]. Multiple signaling pathways have been shown to contribute to the regulation of transcription of the inducible gene in hypoxia-inducible gene and protein stabilization of HIF-1α even in normoxia. Some of these pathways include extracellular signal-regulated Ras/kinase (ERK) and mitogen-activated p38-protein kinase (MAPK), phosphatidylinositol-3-kinase (PI3K) pathway and mTOR signaling pathway [15]. In addition to hypoxia, other oncogenic pathways including signaling pathways for growth factors or genetic loss of tumor suppressor genes, such as VHL and PTEN, over-regulate HIF-1 activity [16]. Importantly, for this work it has been shown that HIF-1α is overexpressed in various types of leukemia, including pediatric ALL and interestingly the high expression of this transcription factor correlates with poor survival [17].

Different types of resistance to cytotoxic agents have been identified, including the involvement of membrane transporters. These transporters are proteins that act as ATP-dependent expulsion pumps, causing a decrease in intracellular concentrations and drug resistance [18,19]. Among the most studied are the multi-drug resistance 1 (MDR1) protein (or gp-170). MDR1 is encoded by the *ABCB1* gene, and constitutively expressed in normal tissue [19]. The overexpression of this protein induces excessive flow and therefore an insufficient intracellular concentration of cytotoxic agents even at maximum doses, which results in a resistance to various chemotherapeutic drugs. Drugs most often used in leukemia treatment include anthracycline, vinca alkaloids, and podophyllines, which are substrates for MDR1. Expression of *ABCB1* messenger RNA is frequently detected in tumors of patients with ovarian cancer, acute myeloid leukemia, and other cancers. Some studies have proposed that expression of *ABCB1* messenger RNA correlates with tumor severity intrinsic to drug resistance after chemotherapy [8,20,21]. It is known that in general, in leukemia, the MDR1 phenotype is generally acquired after the administration of chemotherapeutic agents and is more frequent in ALL and in aggressive carcinomas (e.g., breast and ovarian) [21]. Recent studies have shown that HIF-1α positively regulates the expression of MDR1, which represents a mechanism of chemotherapy resistance in various types of tumors [22]. However, the mechanisms that regulate the expression of MDR1 are not fully elucidated.

The transcription factor Yin-Yang-1 (YY1) is known to play a fundamental role both in normal biological processes such as embryogenesis, differentiation, replication, cell proliferation, and in mechanisms of carcinogenesis, tumor progression and metastasis. It is estimated that more than 7% of vertebrate genes contain binding sites for YY1, which reflects the importance of this transcription factor [23]. YY1 participates in response events to various apoptotic stimuli and has been associated with carcinogenic processes by activating relevant proto-oncogenes such as c-Myc, and downregulating tumor suppressor genes such as p53. An increase in expression and/or activation of this transcription factor has been shown in different neoplasms, such as hematopoietic neoplasia, carcinomas, hepatocarcinoma and retinoblastoma [24,25,26]. Recent studies have shown that YY1 is elevated in patients with non-Hodgkin’s lymphoma and leukemia and its high expression correlates with poor prognosis [27]. Our group recently reported that YY1 regulates the expression of MDR1 and its over-expression is correlated with poor prognosis in ALL pediatric patients [28]. In addition, we demonstrated that high nuclear expression of YY1 correlates with poor survival in leukemia patients. Nevertheless, the role of these transcription factors in the pathogenesis of ALL is not clear and given their possible co-expression and correlation with poor prognosis, it is plausible to think that there is a relationship between these two transcription factors. Our first approach hypothesized that HIF-1α could regulate transcriptionally the expression of YY1 since we previously showed that HIF-1α and YY1 increase their expression under hypoxic conditions [29]. Understanding the regulatory mechanism underlying YY1 expression and its implications in ALL, as well as its relationship with the transcription factor HIF-1α, is important for diagnostic and prognostic purposes.

## 2. Results

### 2.1. Transcriptional Regulation of the YY1 Protein by HIF-1α in Leukemia Cell Lines

Based on independent findings regarding the expression of HIF-1α and YY1 in lymphomas and leukemia [24,30,31], we proposed that there is a correlation between these proteins. To investigate this, we performed a bioinformatics analysis to predict HIF-1α binding sites in the YY1 promoter to described in detail material and methods section. Three putative binding sites located at nucleotides −622 bp, −592 bp and +199 bp were identified with respect to the YY1 gene transcription start site (TSS) (Figure 1A).

To determine if HIF-1α can regulate the expression of YY1 through activation of its promoter, we evaluated the role of each binding site in regulating the promoter region of the gene encoding YY1. The YY1 promoter region was cloned into the reporter plasmid pGL3 as described in the Materials and Methods Section. The reporter plasmid pGL3-YY1-pro-luc was generated. A single or double mutation of the sites in the YY1 promoter was performed. The mutants were designated as pGL3-YY1-MutA-pro-luc (site −622), pGL3-YY1-MutB-pro-luc (site −592) and pGL3-YY1-MutC-pro-luc (site +199) for the single mutants and as pGL3-YY1-MutAB-pro-luc and pGL3-YY1-MutBC-pro-luc for the double mutants. A triple mutant, pGL3-YY1-MutABC-pro-luc was also generated. Reporter plasmids containing their respective mutations were transfected into the PC3 cell line as in the transfection model previously reported [21]. Figure 1B shows the luciferase results. For plasmid pGL3-YY1-MutA-pro-luc and pGL3-YY1-MutC-pro-luc the luciferase/B-galactosidase results were significant at * *p* < 0.005 when comparing with pGL3-YY1-pro-luc plasmid, which contains the complete promotor of YY1. However, the most dramatic effect observed with the reporter gene (luciferase) was obtained with the plasmid pGL3-YY1-MutB-pro-luc (* *p* < 0.001), for which luciferase/β-galactosidase activity was very similar to the results observed with the empty plasmid. This result was corroborated by the luciferase/β-galactosidase activity observed with the plasmid containing double and triple mutant. When sites A, B and C were mutated, the activity of the reporter plasmid was affected, and the fold change with respect to control is shown (Figure 1B). These results show that the sites at −622, +199, and especially site −592 play an important role in the positive regulation of YY1 by HIF-1α.

To confirm the interaction of the transcription factor HIF-1α and the promoter region of YY1, ChIP assays were performed. Chromatin from the RS4;11 cell line was used. For immunoprecipitation, an anti-HIF-1α antibody was used, and then segments were amplified by PCR using specific oligonucleotides for each possible binding site of HIF-1α in the YY1 promoter region. The results are shown in Figure 1C, and we observed that HIF-1α binds to all three sites (+199, −592 and −622) in the YY1 promoter. Non-immunoprecipitated chromatin was used as a positive control, and control IgG was used as a negative control. ChIP is shown as efficiency of the immunoprecipitation YY1 and control (Figure 1C).

### 2.2. Induction or Inhibition of HIF-1α Expression Affects in the YY1 Expression

In order to demonstrate whether the inhibition of HIF-1α can modify YY1 expression, we incubated the leukemia cell line RS4;11 in normoxia and hypoxia conditions for different times. Figure 2A, upper panel shows a representative photomicrograph of HIF-1α immunostaining in RS4;11 cells cultured under normoxia and hypoxia over time (0, 0.5, 1, 3, 6 and 9 h). The results show, as expected, that there is a gradual increase in the expression of HIF-1α over time under hypoxia. Interestingly, immunostaining was preferably observed at the nuclear level indicating activity of this transcription factor. When we quantified the expression of this transcription factor, we observed a significant gradual increase after three hours under hypoxic conditions (*p* = 0.02) until nine hours (Figure 2A, bottom panel). Remarkably, very similar results were obtained when we evaluated YY1 expression in the same experiment; Figure 2B, upper panel shows a representative photomicrograph of YY1 immunostaining. Immunostaining observed mainly at the nuclear level revealed a gradual increase in YY1 expression with respect to time under hypoxic conditions. When we performed the quantification of the expression of this transcription factor, a significant increase was observed from six hours in hypoxic conditions (*p* = 0.03) and a gradual escalation after 9 h (*p* = 0.005) (Figure 2B, bottom panel). The increased expression of YY1 under hypoxic conditions were analyzed by RT-PCR (Figure 2C). A similar significant increase in the expression of YY1 mRNA is observed at 3, 6 and 9 h under hypoxic conditions. We next evaluated the expression of genes targeting these transcription factors. The expression of MDR1 (target gene of HIF-1α) and c-Myc (target gene of YY1) significantly increased after 6 h in hypoxic conditions (Figure 3).

We next explored the effects of inhibiting HIF-1α activity using a chemical inhibitor 2-methoxyestradiol (2ME). Figure 4A shows a representative photomicrograph of HIF-1α and YY1 immunostaining in RS4;11 after a 6-h culture under hypoxia. We found that treating cells with 2ME (0.5 or 1 μM) inhibits the HIF-1α expression. As expected, the expression of YY1 also decreased proportionally with HIF-1α expression. These results were corroborated by real-time PCR analysis under normoxia and hypoxia at 3 and 6 h with 2ME (0.5 μM)(Figure 4B).

We then evaluated if 2ME sensitizes RS4;11 cells to the chemotherapeutic drug, etoposide, since it has been shown that both HIF-1α and YY1 positively regulate the expression of MDR1 [28,32]. Remarkably, we found that the combined treatment of 2ME with etoposide (2ME/Eto) significantly decreases the percent viability of RS4;11 compared with single treatment with 2ME, etoposide or control (*p* = 0.01) (Figure 4C).

### 2.3. Correlation of HIF-1α and YY1 Expression in Pediatric Patients with ALL

We next examined the expression of HIF-1α and YY1 in 108 patients with ALL chemotherapy, untreated, as well as 50 healthy controls; the clinical characteristics of our study population are shown in Table 1. In Figure 5A, we show a representative photomicrograph of immunostained HIF1α and YY1, where both proteins are observed in patients with ALL were compared to healthy controls. The expressions of HIF-1α and YY1 were mainly at the nuclear level. When expression of both proteins was quantified, a significant increase in their expression was observed in patients with ALL as compared to healthy controls, *p* = 0.0001 for HIF-1α and *p* = 0.04 for nuclear YY1 (Student *t*-test) (Figure 5B). In order to validate our results in peripheral blood an analysis of YY1 expression in patients undergoing treatment was conducted. YY1 levels were measured from patients at time of diagnosis as well as different phases of treatment (remission and consolidation) and compared with YY1 expression in healthy controls. Results show that patients at diagnosis have a significantly greater expression of YY1 compared with healthy controls (*p* = 0.001). However, after remission and consolidation phases of treatment, expression of YY1 decreases to levels similar to that of control cells (Figure 5C). We performed an analysis of mRNA expression for HIF-1α and YY1 in 46 bone marrow samples from pediatric patients with ALL. The results show that elevated expression of HIF-1 mRNA correlates with high expression of YY1 mRNA (*r* = 0.35, *p* = 0.0197) (Figure 5D).

### 2.4. Network Analysis of HIF-1α/YY1 and Correlation between HIF-1α and YY1 Expression in ALL

The analysis performed showed the protein–protein interaction (PPI) relationships for YY1 and HIF-1α genes. With a combined score of >0.4 when selected and restricted to *Homo sapiens*, we found a close relationship between seven genes confirmed by curated and experimental data. The analysis identified the co-activator EP300 as likely playing a role in the regulation of HIF-1α and YY1, so YY1 probably also plays an important role in the regulation of EP300 and HIF-1α. Additionally, the meta-analysis showed an interaction with HIF1AN, the inhibitor of HIF-1α, VHL, a protein involved in the ubiquitination and degradation of HIF-1α, TCEB1, a subunit of the transcription factor B (SIII) complex closely regulated by VHL and finally with EGLN1, which codes a protein that catalyzes the post-translational formation of 4-hydroxyproline in HIF-1α proteins (Figure 6A). In order to corroborate our data, we performed bioinformatics analysis. The microarray data were from 87 ALL-B phenotype samples out of the 127 different types of leukemia samples present in a related data set (GSE7186). Figure 6B showed the selective analysis of HIF-1α and YY1 co-expression, where significant correlation was found for ALL-B samples (** *p* < 0.0011, *r* = 0.3597). These results are consistent with our results showed in Figure 1 where we demonstrated that HIF-1α expression and YY1 expression are positively correlated in samples derived from pediatric ALL patients.

## 3. Discussion

HIF-1 was initially characterized as a molecular regulator to control oxygen homeostasis [35]. HIFs have long been studied in solid tumors, where they have been shown to promote wide-ranging tumor-promoting processes, including neo-angiogenesis, cancer metabolism, maintenance of cancer stem cells, and immune evasion [36]. Tumor hypoxia strongly correlates with poor prognosis in solid cancers [37], and in the past 10 years, clear involvements of HIFs have also been demonstrated in leukemia. However, the role of HIF-1α in the transcriptional regulation of the YY1 gene so far remains largely unknown. Upon analysis of the YY1 promoter, we found three putative HIF-1α binding sites. We therefore hypothesized that HIF-1α is involved in positively regulating YY1 and is the reason for chemoresistance in tumor cells.

To demonstrate the above, we performed reporter plasmids and site-directed mutagenesis and eliminated each of the binding sites for HIF-1α. The results demonstrated that mutation of the −592 site results in a drastic decrease in the activity of the reporter gene (site with higher 9.3. JASPER), indicating that this site is the most important for the regulation of the expression of YY1 by HIF-1α. Furthermore, ChIP assays demonstrate that the binding of HIF-1α in the YY1 promoter in ALL cells. To corroborate the impact of HIF-1α inhibition of YY1 positive expression, we cultured RS4;11 leukemia cells in hypoxic conditions over time. The results of this experiment reveal that hypoxia induces significant nuclear translocation of HIF-1α after two hours of hypoxia. Importantly, in the same experiment the expression of YY1 was also upregulated under 6 h of hypoxia. This was corroborated by WB analysis. To demonstrate the activity of HIF-1α and YY1, we evaluated the expression of the target genes of these transcription factors, MDR1, regulated by HIF-1α and YY1 [22], and c-Myc regulated by YY1 [38]. The results demonstrate that hypoxic conditions induce the activity of both HIF-1α and YY1. These results are very significant, if we consider that tumor cells show an increase HIF-1α activity (due to either conditions of hypoxia or the presence of a pro-inflammatory environment) and that increase correlates with more angiogenesis, resistance to apoptosis and induction of MDR1 expression, among other mechanisms [22,39]. These findings represent a new mechanism induced by HIF-1α to increase the malignancy of tumor cells. In addition, we recently published that leukemia cell RS4;11 co-expressed both with HIF-1α and YY1 under hypoxia, which correlated with a downregulation of Fas expression. During hypoxia, the levels of apoptosis diminished after an agonist of FasL (DX2) treatment. Moreover, a bioinformatics analysis revealed that patients with high levels of HIF-1α also express high levels of YY1 and low levels of Fas. These results suggest that YY1 negatively regulates the expression of the Fas receptor, which would be involved in the escape of leukemic cell from the immune response is another mechanism to contributing to the ALL pathogenesis [29]. Once we demonstrated that HIF-1α positively regulates the expression of YY1, we then evaluated the effect of using a chemical inhibitor of HIF-1α activated activity. A natural metabolite of 17-β-estradiol that does not bind to the estrogen receptor, 2-methoxyestradiol (2ME), has anti-proliferative and anti-angiogenic activity [40,41]. Therefore, 2ME was approved several years ago by the FDA for use in humans to treat cancer, and to date it is used in patients with nasopharyngeal cancer, multiple myeloma, prostate cancer, among others [41,42]. However, its effect on ALL cells has not been studied. To evaluate the effect of 2ME in tumor cells, we performed a series of experiments using this chemical inhibitor to treat RS4;11 cells. As previously demonstrated, RS4;11 cells under 6 h of hypoxic conditions show a significant increase in HIF-1α and YY1 expression. In order to discern the localization of HIF-1α and YY1 expression in ALL cells before and after treatment with 2ME, we performed immunocytochemical assays. The results demonstrated a clear decrease in HIF-1α and YY1 expression after treatment with this chemical inhibitor, and very similar results were obtained after performing RT-PCR.

Next, it was important to evaluate whether treatment with 2ME induces reversal of chemoresistance in ALL cells, so we performed cell viability tests with RS4;11 cells after treatment with 2ME alone and in combination with different concentrations of the chemotherapeutic drug etoposide, which is used in the treatment of pediatric patients with ALL. The results are shown in Figure 4C. A sensitivity to the drug of up to 40% after treatment with 2ME is observed in contrast to cells treated with the drug or inhibitor separately. This is very relevant, as this drug is currently used in the treatment of pediatric patients with ALL, and is a substrate of MDR1 [43]. These results are consistent with those reported by other groups demonstrating that 2ME sensitizes different cancer cells to die in the presence of chemotherapeutic drugs, including acute myeloblastic leukemia [44].

To corroborate our results in vitro, we evaluated HIF-1α and YY1 expression in patients with ALL. Our results demonstrated the significant expression of both proteins in ALL pediatric patients. Furthermore, this expression was directly proportional. These results are novel, considering that no previous studies have shown the involvement of HIF-1α in the regulation of YY1 in the pathophysiology of ALL. Our results clearly show constitutive expression of both proteins, strongly suggesting the importance of YY1 protein expression in the pathogenesis of ALL pediatric patients. In this study, we determined HIF-1α and YY1 expression levels by ICC and analyzed this expression in mononuclear peripheral blood cells and bone marrow cells derived from pediatric ALL patients. Our results are consistent with previous findings which demonstrate that tumor hypoxia strongly correlates with poor prognosis of solid cancers [37]. Additionally, in the past 10 years, clear evidence of HIFs have been shown to be involved in the pathogenesis of leukemia [17,44,45]. However, some reports have suggested that HIFs may exert tumor suppressive functions in acute myeloid leukemia, albeit this may be limited to specific disease sub-contexts. For example, Vukovic et al., demonstrate that knockdown of HIF-1α or HIF-2α in human acute myeloid leukemia (AML) samples results in their apoptosis and inability to engraft. Both Hif-1α and Hif-2α synergize to suppress the development of AML. (Vukovic M, Hif-1α and Hif-2α synergize to suppress AML development but are dispensable for disease maintenance [45].

It has been shown that YY1 is an important negative regulator of the tumor suppressor factor p53 [46]. Wu S. et al. demonstrated that inhibition of YY1 reduced the accumulation of HIF-1 α and its activity under hypoxic conditions, and consequently downregulated the expression of HIF-1 α target genes. In addition, it was demonstrated that the downregulation of HIF-1 α by inhibiting YY1 is p53-independent. Therefore, YY1 inhibition could be considered as a potential tumor therapeutic strategy to give consistent clinical outcomesYY1 is associated with HIF-1 α regulation under hypoxia, and targeting YY1 might be a potential therapeutic strategy of solid cancer [47].

Very limited studies are available on the function of HIF1 factors in ALL. So far, it has been shown that HIF-1α is expressed in ALL that reside in the BM [48]. Accordingly, HIF-1α is induced by stroma-mediated AKT/mTOR signaling in pre-B-ALL, and confers resistance to chemotherapy [49]. This information is consistent with our findings which show that HIF-1α positively regulates YY1 expression, and that several transcription factors are reportedly involved in chemoresistance of different types of cancer including ALL [10,11,33,34]. These findings represent a new mechanism of chemoresistance in ALL. In addition, Zhea, N. et al. demonstrated a high expression of HIF-1α in human AML cell lines and the inhibition of HIF-1α by 2ME has potential anti-leukemia activity through activation of the mitochondrial apoptotic pathway mediated by ROS. In addition, it is not cytotoxic to normal cells. 2ME is therefore a potential candidate for the treatment of AML [44]. Our results, demonstrating chemo-sensitization of drugs in leukemia cells using 2ME, strongly suggest that this metabolite can also be effective in ALL.

Based on data retrieved from Oncomine, we found that HIF-1α and YY1 mRNA were expressed in several leukemia subtypes, especially ALL, as shown in Figure 6. We found a positive correlation between the expression of HIF-1α and YY1 in several data sets analyzed from the leukemia study by Anderson et al. [34]. This correlation is consistent with findings from our in vivo patient samples and confirm the interaction and regulation of YY1 by HIF-1α and also are consistent with our previously reports to reveal that by a bioinformatics analysis that patients with high levels of HIF-1α also express high levels of YY1 and low levels of Fas [29]. These findings indicate that HIF-1α and YY1 might participate in the initiation and progression of ALL via positive transcriptional regulation. On the other hand it has been described that the inhibition of YY1 disrupts hypoxia-stimulated HIF-1α stabilization in a p53-independent manner, reducing the accumulation of HIF-1α and its activity under hypoxic condition, and consequently downregulated the expression of HIF-1α target genes [47]. Therefore, the transcriptional regulation of YY1 on the HIF-1a promoter is feasible, and possible participates in the regulation of its expression, which suggests a bidirectional regulation.

A bioinformatic analysis with Cytoscape permits the identification of active subsets/modules. A network was analyzed in conjunction with gene expression databases (microarray databases used in this study: ONCOMINE, GEO-NCBI) to identify sets of connecting interactions between proteins by identifying interaction subsets in which genes show particularly high levels of differential expression. The interactions contained within each subset provide hypotheses for regulatory and signaling interactions controlling observed changes in expression. One can search for groups (highly interconnected regions) and load any network in Cytoscape. Depending on the type of network, groups can have different meanings. Networks are designed with automated algorithms. Our Cytoscape analysis identified interactions between HIF-1α and YY1, and this correlation was confirmed by experimental findings obtained with ChIP and binding site mutation. The analysis identified co-activator EP300 as playing a role in the regulation of HIF-1α and probably in YY1 as well, but interestingly, HIF-1α may also play an important role in the regulation of both EP300 and YY1. Additionally, the meta-analysis showed an interaction with HIF1AN, the inhibitor of HIF-1α, VHL, a protein involved in the ubiquitination and degradation of HIF-1α, TCEB1, a subunit of the transcription factor B (SIII) complex closely regulated by VHL and finally EGLN1, which catalyzes the post-translational formation of 4-hydroxyproline in HIF-1α proteins, from data that was published already by other research groups [50].

This is the first report describing a correlation between HIF-1α and YY1 expression in pediatric ALL patients, and this study identifies HIF-1α and YY1 as potential disease markers, which could be considered biomarkers at the time of diagnosis for predicting disease behavior. We also propose that the use of pharmacological or chemical inhibitors targeting HIF-1α and YY1 could be an alternative treatment for pediatric patients with ALL that are known to be positive for HIF-1α and YY1 expression, thus offering a therapeutic alternative for this disease.

The model in Figure 7 summarizes all our findings.

## 4. Materials and Methods

### 4.1. Ethical and Biosecurity Aspects

This project involved the manipulation of blood of pediatric patients with ALL. Due to this, during development, the standards of good laboratory practices were followed to avoid occupational risks. The study was in accordance with the regulations of the General Health Law on Research in Mexico, as well as the research standards of the Hospital Infantil de México Federico Gómez. A letter of informed consent was signed in all cases. The data provided by the patient’s clinical file were kept confidential, according to the Helsinski international standard for research.

### 4.2. Patients

In this study, mononuclear cells from peripheral blood (PBMC) samples were isolated from 108 patients with ALL (0–16 years of age) who were diagnosed between 2009 and 2018. 10 samples were collected during the different phases of treatment (remission and consolidation). The samples were obtained from the Oncology Pediatric Services of both the Hospital Infantil de México Federico Gomez and the Hospital Pediatrico Moctezuma. In this study, we also included 46 bone marrow (BM) samples from pediatric patients with ALL, who were diagnosed between 2014 and 2016 in the Oncology Pediatric Service of the Instituto Nacional de Pediatria, SSA. The diagnosis was established by cytological examination of bone marrow smears according to the French-American-British (FAB) group. Cytochemical tests included staining for Periodic Acid Schiff (PAS) and myeloperoxidase (MPO) [51,52]. The pediatric control group was composed of 50 children (mean age: 7.8 years; range: 5 to 15 years) who were admitted to the hospital for elective surgery and who were free of any known viral or bacterial infections.

### 4.3. Cell Culture

Cells were cultured from the RS4;11 (acute phenotype B lymphoblastic leukemia cell line, ATCC: CRL-1873) and PC3 cell lines (prostatic carcinoma; ATCC CRL-1435), which overexpress YY1. A 25 cm^2^ box was cultured with RPMI advanced 1640 medium supplemented with 5% FBS (GIBCO-Invitrogen), 1% *v*/*v* L-glutamine (GIBCO-Invitrogen), 1% *v*/*v* sodium pyruvate (100 mM GIBCO-Invitrogen), 1% antibiotic (GIBCO-Invitrogen) and 1% non-essential amino acids (GIBCO-Invitrogen). Cells were maintained in culture at 37 °C and 5% CO_2_.

### 4.4. Determination of the Putative Binding Sites of HIF1α in the YY1 Promoter

The prediction of the binding sites for HIF-1α in the promoter of the YY1 gene was performed by the TESS (Transcription Element Search System) program which conjugates the databases of TRANSFAC v6.0, JASPAR 20060301, IMD v1.1 and IWC/GibbsMat v1. 2000 nucleotides upstream (−2000 bp) of the gene promoter ATG sequence at 350 nucleotides downstream (+350 bp) were analyzed. Three putative binding sites located at nucleotides −622 bp, −592 bp and +199 bp were identified with respect to the YY1 gene TSS.

### 4.5. Cloning of the Promoter Region of YY1

Genomic DNA was extracted from a healthy volunteer using TRIzol (invitrogen) and used as template to amplify the promoter region of the YY1 (−2000 bp to +350 bp relative to the start site) by PCR using a specific set of primers. Once amplified, the YY1 promoter was purified and cloned into the vector pJet (Thermo Fisher Scientific). Then the YY1 promoter was subclonated from pJet to the pGL3 vector (Promega) generating the construct pGL3-YY1-pro with Luciferase as a reporter gene.

### 4.6. Site-Direct Mutagenesis in Putative Binding Sites for HIF-1α of the Promoter YY1

Once the pGL3-YY1-pro construct was obtained, a site directed mutagenesis was performed in the three putative binding sites using a system of commercial site-directed mutagenesis QuickChange Lighting Site-Directed Mutagenesis, (Agilent Technologies, Santa Clara, CA, USA) following the manufactures instructions. Importantly, the computer algorithm used provided the supplier with the primer design. The incorporation of the mutations in the putative HIF-1α sites was performed by PCR using the primers show in Table 1. After the restriction enzyme digestion of the PCR products with Dpn-1, *E. coli* strain DH5-alpha bacteria were transformed with the digested PCR products.

### 4.7. Transfection of Cell Lines with the Construct Generated

PC3 cells were co-transfected with 2 μg of DNA total from each of the plasmid constructs generated (pGL3-YY1-pro or the mutants generated from it) and pCMV-SPORT-β-gal vector (Invitrogen, Waltham, MA USA) at a 6:1 proportion. The pCMV-SPORT-β-gal vector was used to normalize the enzymatic activities from the reporter genes regarding transfection efficiency. Following the manufacturer’s recommendations, Lipofectamine 2000 (Invitrogen) was used as a transfection reagent. After 48 h post-transfection, intracellular proteins were obtained to determine the enzymatic activity of the implicated reporters. Using commercial substrates, the luciferase (Promega, Madison, WI, USA), and β-galactosidase (Clontech, Santa Clara, CA, USA), activities were measured in the multimodal reader plates EnSpire (Perkin Elmer, Waltham, MA, USA).

### 4.8. Chromatin Immunoprecipitation

RS4;11 cells (1×10^6^ previously stimulated using hypoxic or normoxic conditions (at the indicated time points) and then chromatin immunoprecipitation (ChIP) was performed as previously described [28]. The specific antibody against HIF1α (ab2185; Abcam) was used. The DNA recovered after the ChIP assay was used as template for PCR reactions using the specific set of primers (Table 2).

### 4.9. Treatments and Exposure of ALL Cells to Hypoxic Conditions

RS4;11 (5 × 10^5^) cells in culture were exposed to low oxygen conditions (1% of O_2_) in a hypoxia chamber Bactrox (Scientific Biogen, Madrid, Spain), during different periods. Subsequently the cells were harvested to perform immunocytochemistry and RT-PCR assays, cells were treated or untreated for 3 or 6 h with 2ME (0.5 or 1 µM) under normoxia or hypoxia. The RS4;11 cells were pretreated 12 h with 2ME (0.5 µM) and then treated or untreated for 18 h with etoposide (0.0625 μM). The viability of RS4;11cells were evaluated after treatment as describe below.

### 4.10. Immunocytochemistry

Immunostaining was performed as previously describe [28]. Anti-HIF-1α, anti-c-Myc, anti-YY1 or anti-MDR1 (Novus Biological, Litletown, CO, USA).

### 4.11. Real-Time RT-PCR Assays

For YY1 mRNA expression, the total RNA was obtained from RS4;11, 5 × 10^6^ cells were cultured under normoxic or hypoxic conditions. After treatment, cells were collected and RNA was purified using the miRNeasy Mini Kit (QIAGEN, Germantown, MD, USA) according to manufacture instructions. cDNA was performed using 1 µg of RNA of each sample according to TaqMan Reverse Transcription kit (Applied Biosystems, Foster City, CA, USA). Gene expression was analyzed by real-time PCR using Maxima SYBR Green/Fluorescein qPCR Master Mix (Thermo Scientific, Waltham, MA, USA) according to manufacture instructions. In addition, the total RNA was obtained from bone marrow aspirate samples from patients with ALL and healthy controls as previously described [52]. Subsequently, the expression of HIF-1α and YY1 was determined using the Universal Probe Library Set, Human (Roche Diagnostics GmbH, Mannheim, Germany). The expression levels were then in a computer attached to the thermocycler StepOne™ Real-Time PCR System (Applied Biosystems Foster City, CA, USA). Data capture and analysis was carried out with the thermal cycler program. The PCR conditions were 1 cycle at 95 °C/10 min, 45 cycles of 95 °C/15 s and 60 °C/1 min. Using specific primers for HIF1α, and YY1. To normalize the amount of cDNA the OAZ gene (ornithine decarboxylase antizyme) was used. The sequence of the primers for each gene are shown in Table 2.

### 4.12. Cell Viability Assays

Cell viability assays were performed via MTT colorimetric assays following the manufacture’s instruction (Roche Diagnostics Co., (Basel, Switzerland). Briefly, after RS4;11 cells treatment the MTT reagent was added, after 4 h the solubilization of the salts was carried out by means of the solubilizing agent included in the MTT kit to later incubate for one night. Once the incubation occurred, the plate is revealed by quantifying the absorbance in a multi-reader EnSpire (PerkinElmer) plate of PE at 620 nm.

### 4.13. Network Analysis of HIF-1α/YY1 and the Construction of Gene Networks Related to Function

To demonstrate the functional interactions between HIF-1α and YY1 and other transcriptional factors, a bioinformatics analysis was performed with GeneMANIA using the free software Cytoscape 2.8, which visualizes biological networks and integrates data, and the database Oncomine [53]. Typically, annotations used by Cytoscape correspond to the GEO database (Gene Ontology Database) [54]. Both databases permit free access to microarray banks and data meta-analyses and networks, which allow the prediction of interactions between genes or proteins of interest.

To elucidate the interactions between YY1/HIF-1α and other proteins, a bioinformatics analysis was performed with The Search Tool for the Retrieval of Interacting Genes (STRING) database (https://string-db.org/), which visualizes biological networks and integrates data from curated and experimental determinations [32].

### 4.14. Bioinformatics Analysis of and Correlation between HIF-1α and YY1 Gene Expression in ALL

An analysis of HIF-1α and YY1 expression levels in ALL was performed using a public data set of microarrays retrieved from the Oncomine and Gene Expression Omnibus databases, derived from a published analysis reported by Andersson, A. et al. [48].

### 4.15. Statistical Analysis

A database was analyzed, and the information was processed using a statistical analysis program (Prism 4^®^ from GraphPad Software, Inc., San Diego, CA, USA), and the evaluation of the difference in the number of positive cells from the immunocytochemical reactions was performed by analysis of variance (ANOVA). The correlation analysis was performed using Pearson analysis. All the data in the graphs are represented as a mean ± SEM. A value less than or equal to 0.05 was considered significant. For statistical analysis we used the STATA program (version 11.00).

## 5. Conclusions

In conclusion, our findings demonstrate for the first time that HIF-1α regulates transcriptional YY1 and increasing or inhibiting HIF-1α expression directly affects YY1 expression in ALL cells. Furthermore, HIF-1α and YY1 expression are increased in pediatric patients with ALL, and high expression of HIF-1α correlates with the presence of YY1. Therefore, both HIF-1α and YY1 may be possible therapeutic biomarkers in ALL. The clinical significance of this finding across different cancer types has yet to be determined.

## Figures and Tables

**Figure 1 ijms-23-01728-f001:**
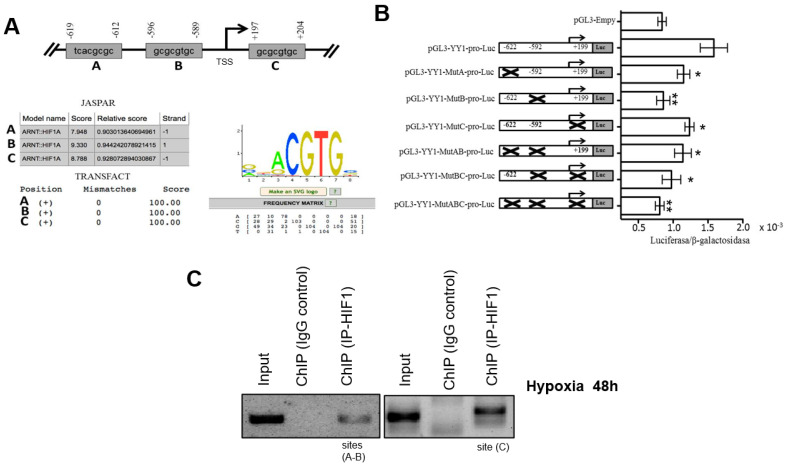
HIF-1α regulates YY1 transcriptional activation by direct interaction with its promotor. (**A**) Three potential binding sites for the transcription factor HIF-1α obtained after bioinformatics analysis using two online servers, JASPAR and TRANSFACT, are displayed. The region from −2000 to +350 bp in the YY1 gene was analyzed for Transcription Start Site (TSS). A weight matrix obtained from the JASPAR database for the transcription factor HIF-1α is displayed. (**B**) Putative binding HIF-1α sites in the YY1 promoter that are involved in regulating expression. Transfection assays were performed using the PC3 cell line to assess the effects of directed mutagenesis at each of the YY1 binding sequences, located at sites −622 bp, −592 bp and +199 in the promoter region of the YY1 gene. The schematic shows each of the mutated sites, and the graph indicates normalized luciferase reporter gene expression levels obtained by measuring β-galactosidase via co-transfection with a reporter gene plasmid; fold changes are reported. The results are representative of three independent experiments (one-way ANOVA, * *p* < 0.005, ** *p* < 0.001). (**C**) ChIP was conducted for each potential HIF-1α binding site in the YY1 promoter. The results show that HIF-1α binds the promoter region of YY1. Results of three independent experiments are shown.

**Figure 2 ijms-23-01728-f002:**
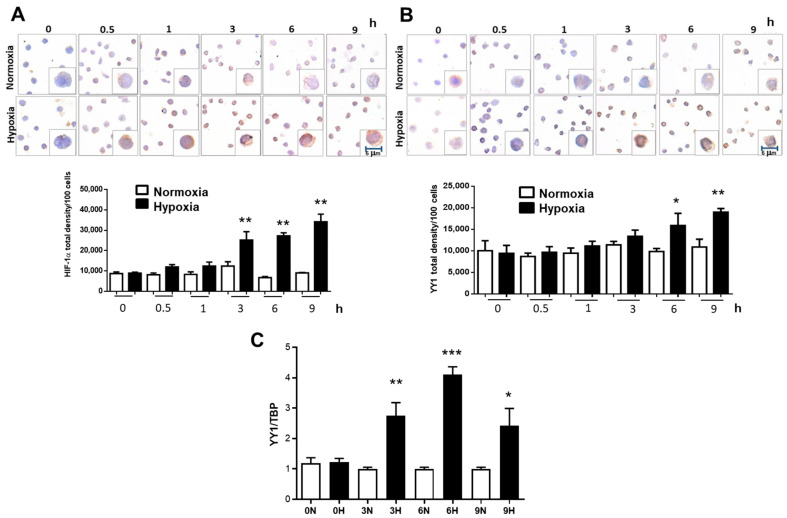
Hypoxia induces expression of YY1. RS4:11 were cultured under normoxia or hypoxia conditions for different times (0, 0.5, 1, 3, 6, 9 h). Slides were then prepared for ICC staining for HIF-1α, (**A**) and YY1 (**B**). The immunostaining shows significant increase of HIF-1α after two hours (* *p* = 0.035) under hypoxia compared with normoxia conditions. Interestingly, YY1 expression increases significantly later, at time 6 h (***p* ≤ 0.03), both proteins continue to increase after 9 h (*** *p* = 0.005 one-way ANOVA-test). (**C**) Analysis of YY1 expression by evaluated using real-time PCR. Bars represent the media of an assay by triplicate. Differences were analyzed by one-way ANOVA and Tukey’s multiple comparisons test (** *p* = 0.01, *** *p* = 0.0005, * *p* = 0.05 one-way ANOVA-test).

**Figure 3 ijms-23-01728-f003:**
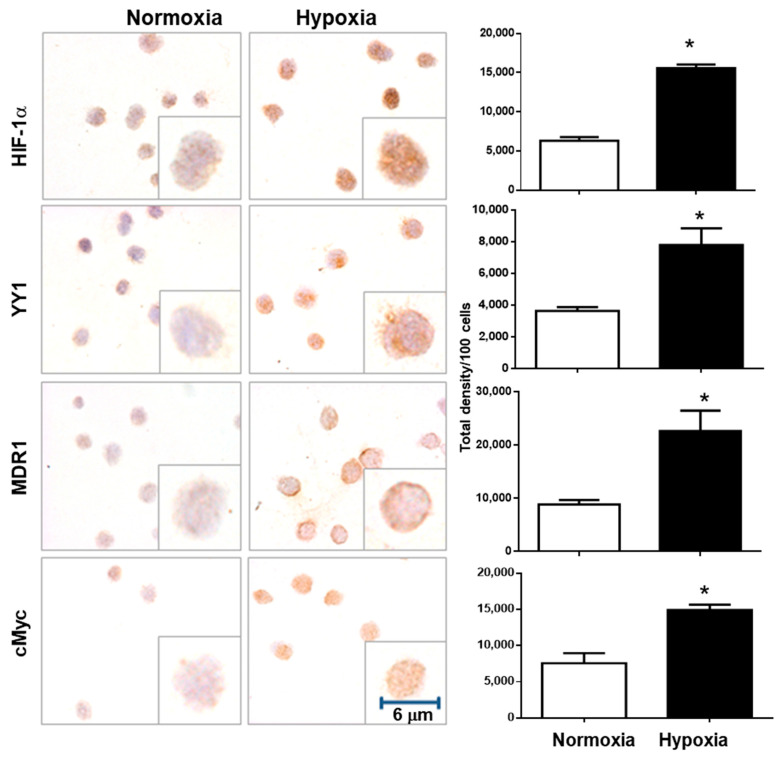
Hypoxia induces the expression of YY1 target genes. To evaluate the transcription factor activation of HIF-1α and YY1, ICC for target genes of each transcription factor were performed. MDR1 for HIF-1α and c-Myc for YY1. Significant increases for both target genes were observed after 6 h. (* *p* < 0.05 Student *t*-test). Representative images of a triplicate of each experiment are shown.

**Figure 4 ijms-23-01728-f004:**
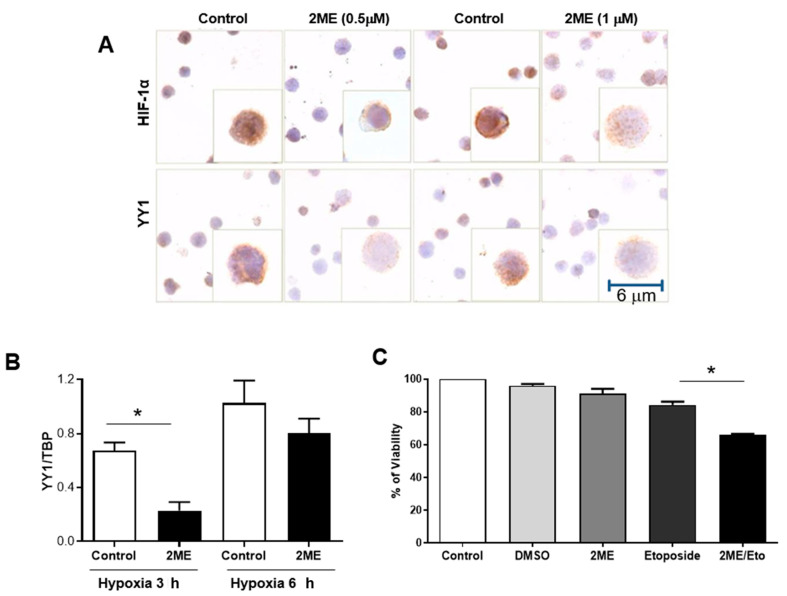
The abrogation of HIF-1α by a chemical inhibitor induces a reduction in YY1 expression. (**A**) is an ICC staining of HIF-1α and YY1 in RS4;11 cells, untreated and treated with 2ME (0.5 and 1 μM) for 6 h. Results show that when HIF-1α is inhibited with 2ME, YY1 decreases correspondingly. (**B**) These results were corroborated by real-time PCR, untreated and treated with 2ME (0.5 μM) for 3 or 6 h. A significant decrease was found at 3 h under hypoxia (* *p* = 0.01 one-way *t*-student). At 6 h there were no significant changes but there was a clear trend. (**C**) RS4;11 cells were treated with 2ME alongside with etoposide, a chemotherapy drug. A significant decrease in viability of RS4;11 cells is observed (* *p* = 0.01 one-way ANOVA-test) when treated with etoposide in combination with 2ME. Graph shows the results of triplicates of three independent experiments. RS4;11 cells were pre-treated for 12 h with 2ME (0.5 µM). Subsequently, the cells were treated with etoposide (0.125 µg/mL). Cell viability was determined after 50 h of co-treatment using MTT (* *p* ˂ 0.05, vehicle or Eto vs. 2ME + Eto).

**Figure 5 ijms-23-01728-f005:**
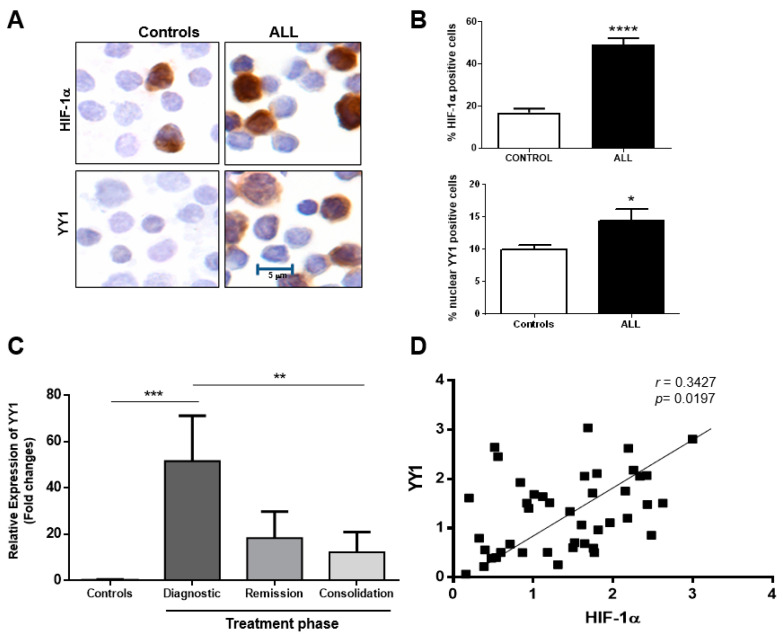
Overexpression of HIF-1α and YY1 in samples from ALL pediatric patients. (**A**) Representative microphotography that show the ICC staining for HIF-1α and YY1 in peripheral blood cells from ALL pediatric patients which clearly shows an overexpression of both proteins in ALL patients as compared with healthy controls. (**B**) This difference is significant when 108 ALL patients are evaluated (**** *p* = 0.0001 and * *p* = 0.04, HIF-1α and YY1, respectively; Student *t*-test). (**C**) RT-PCR of YY1 levels measured from patients at diagnoses, remission and consolidation shows that YY1 levels are significantly greater in 10 ALL patients at time of diagnoses compared to healthy controls (*** *p* = 0.001) but decrease to levels similar to those of healthy control during remission and consolidation phases of treatment (** *p* = 0.01 one-way ANOVA test). (**D**) To corroborate results from peripheral blood cells, RT-PCR was performed for bone marrow samples of 46 pediatric patients with ALL. The results again show a strong positive correlation between HIF-1α and YY1 (* *p* = 0.0197, *r* = 0.35, Pearson analysis).

**Figure 6 ijms-23-01728-f006:**
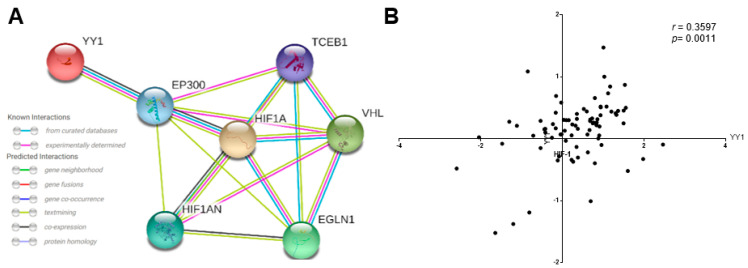
Gene expression and correlation of HIF-1α and YY1 in ALL. (**A**) Network analysis between HIF-1α/YY1. The Search Tool for the Retrieval of Interacting Genes (STRING) database [32], was used to visualize biological networks and integrate data of the protein–protein interaction between HIF-1α and YY1 [33]. The version 11.0 of STRING (Academic Consortium) was employed to seek for the protein–protein interaction (PPI), data limited to *Homo Sapiens* and a confidence score >0.4. (**B**) Analysis of HIF-1α and YY1 expression levels in several subtypes of ALL was performed using a public dataset of microarrays retrieved from the Oncomine database and the Gene Expression Omnibus NCBI gene expression and hybridization array data repository, obtained from an analysis from Anderson et al. [34]. According with Oncomine^TM^ the results shown that HIF-1α expression levels correlated with YY1 expression (*** *p* < 0.0001, *r* = 0.468, Pearson Analysis).

**Figure 7 ijms-23-01728-f007:**
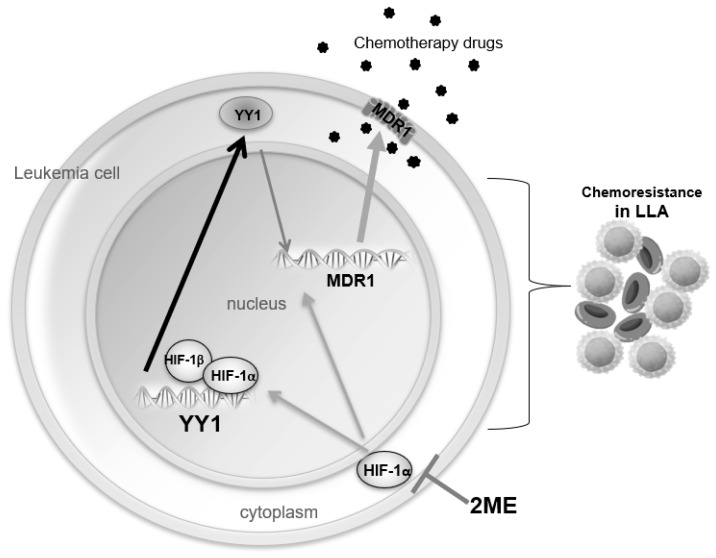
A schematic of HIF-1α transcriptionally regulating YY1 expression. In this study we demonstrated that HIF-1α transcriptionally regulates YY1. In addition, it has been shown that both HIF-1α and YY1 increase expression of MDR1, a membranous protein which pumps out chemotherapy drugs, inducing chemoresistance in leukemia cells. Interestingly, here we demonstrated that this pathway is inactivated by the inhibition of HIF-1α by 2ME, which induces sensitizes leukemia cells to chemotherapy.

**Table 1 ijms-23-01728-t001:** Clinical patient characteristics.

Total Number	108
Gender	
Female	45
Male	63
Age (years)	7.8 (0.1–16)
Phenotype	
B	32
Pre-B	40
Pro-B	13
T	11
Unknown	12
Survivor	79
Deaths	29

**Table 2 ijms-23-01728-t002:** Primer sequences.

Name	Sequence
YY1pro_Sen (BglI underlined)	CG**AGATCT**GCTTTTTTGAACAGAGAGCC
YY1Pro_Ant (BamH1 underlined)	GA**GGATCC**GGGTGCAAACCG
YY1mutAant (Mutation in bold)	GCCCGCGGCG**AAGA**ACGTCAGCGCGCCGCCGCC
YY1mutAsen (Mutation in bold)	GGCGGCGGCGCGCTGACGT**TCTT**CGCCGCGGGC
YY1mutBant (Mutation in bold)	GGCGGGGCGGCTCG**AGAA**CGCCCTGGCTGGCC
YY1mutBsen (Mutation in bold)	GGCCAGCCAGGGCG**TTCT**CGAGCCGCCCCGCC
YY1mutCant (Mutation in bold)	GTGGTGGTGGCCGG**AAGA**CCCGTGGCCGCCCC
YY1mutCsen (Mutation in bold)	GGGGCGGCCACGGG**TCTT**CCGGCCACCACCAC
Ch(PromYY)_HIFABsen	TTTTGTGGCTGTTGCACCG
Ch(PromYY)_HIFABant	AATCGATCTGTCCGCTGGC
Ch(PromYY)_HIFCsen	AGACCATCGAGACCACAGTGGTGG
Ch(PromYY)_HIFant	CTGCAGAGCGATCATGGGCG
Ch(PromTel)_HIFposSen	AGCGCTGCGTCCTGCT
Ch(PromTel)_HIFposAnt	AGCACCTCGCGGTAGTGG

## Data Availability

The data underlying this article will be shared on reasonable request from the corresponding author.

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
