# Peer review of "Transcriptional Regulation of Yin-Yang 1 Expression through the Hypoxia Inducible Factor-1 in Pediatric Acute Lymphoblastic Leukemia"

_ijms, 2022, doi:10.3390/ijms23031728_

Round 1

Reviewer 1 Report

Major

YY1 has been shown to regulate HIF-1alpha. The authors fail to mention this relation. It is possible that there is bidirectional regulation and this needs to be investigated/addressed.

The correlation between HIF1a and YY1 is not convincing, especially in Fig 5D.

Minor

Several typos

Line 113 promoter spelling

            Describe to change to described

Line 283 “corroborated our date”                  change to “corroborate our data”

Line 377 “It is having been” change to “it has been”

Fig. 7, unclear about the translocation of YY1 into the cytoplasm.

Discussion can be made more succinct by not repeating the results.

Author Response

Answer to the Reviewer 1 Comments Manuscript ID ijms-1460126
Reviewer: 1

  1. a) YY1 has been shown to regulate HIF-alpha. The authors fail to mention this relation. It is possible that there is bidirectional regulation and this needs to be investigated/addressed

Author response a: Thank you for your comment, in this regard, it has been described that the inhibition of YY1 disrupts hypoxia-stimulated HIF-1α stabilization in a p53-independent manner, reduced the accumulation of HIF-1α and its activity under hypoxic condition, and consequently downregulated the expression of HIF-1α target genes (DOI: 10.1158/0008-5472.CAN-12-0366). On the other hand, the analysis of the HIF-1 a promoter shows YY1 binding sites. A bioinformatic analysis was performed in JASPAR (https://jaspar.genereg.net/) and Eukariotyk Promoter Database (https://epd.epfl.ch/). In the eukariotyk promoter database we use the “Search Motif Tool” and the result indicates 4 potential binding sites for YY1 in the HIF1a promoter from -2000 to 100bp, (YY1 [p-value = 0.001]: -1991, -1657, -1152, -892). In JASPAR we scan the HIF1A promoter from 2000 to 100 bp and find at least 7 potential binding sites for YY1 with a score over 7. Therefore, it is suggested that the transcriptional regulation of YY1 on the HIF-1a promoter is feasible, and participates in the regulation of its expression, which suggests a bidirectional regulation. We added this information in the discussion section of the new version of the manuscript

  1. b) The correlation between HIF1-a and YY1 is not convincing, especially in Fig 5D.

Author response b: A Pearson´s R statistical analysis was performed using GraphPad Prism, the results indicate a significative Pearson´s R value of 0.3427 with a p value: 0.01. This indicates a positive correlation.

  1. c) Line 113 promoter spelling. Describe to change to described

Author response c: We make the suggested change in the text

  1. d) Line 283 corroborates our date. Change to “corroborate our data”

Author response d: We make the suggested change in the text

  1. e) Line 377 “it is having been” change to “it has been”

Author response e: We make the suggested change in the text

  1. f) Figure 7, unclear about the translocation of YY1 into the cytoplasm. Discussion can be made more succinct by not repeating the results.

Author response f: We modified the figure legend in order for it to be more clear

Reviewer 2 Report

The article is concerned around evaluating Yin-Yang transcription factor 1 (YY1) and hypoxia-1 inducible transcription factor (HIF-1) in pediatric ALL cases.

The study is well designed and conducted; I have no objections to the methodology and statistical analysis. Although there are several minor issues that need improvement.

First of all every abbreviation should be defined at first use, even in the abstract – HIF-1 and ALL is missing the explanation.

There is no need to explain an abbreviation more than once through the manuscript – e.g. lines: 100,

Line 70 and below: the human genes should be written in capitals and italicized, the MDR1 gene should be denoted as ABCB1 as for HUGO Gene Nomenclature Committee, changes should be made through manuscript.

 Line 93 – different neoplasias? -  neoplasms

Line 374 – can you state which reports exactly?

Line 494 – table 2?

I highly recommend a revision of the text by a native English language speaker since there are some parts that present several language problems.

Author Response

Answer to the Reviewer Comments Manuscript ID ijms-1460126

Reviewer: 2
1)  First of all every abbreviation should be defined at firs use, even in the abstract –HIF-1 and ALL is missing the explanation.

Author response 1: Thank you for your comments. We included the changes in the text2) There is no need to explain an abbreviation more than once through the manuscrip –e.g. lines: 100.
Author response 2: We make the suggested change in the text
3) Line 70 and below: the human genes should be written in capitals and italicized, the MDR1 gene should be denoted as ABCB1 as for HUGO Gene Nomenclature Committee, changes should be made through manuscript.

Author response 3: We make the suggested change in the text.
4) Line 93- Different neoplasias)- neoplasms

Author response 4: We made the change in the text.
5) Line 374 - can you state which reports exactly?

Author response 5: Evidence that demonstrate that HIF-1α is involved in the pathogenesis of leukemia is described in references 17,40,46. In the other hand, the evidence that shows that HIF are involved in the suppression of the development of leukemia is described in reference [DOI: 10.1084/jem.20150452].

  1. Line 494 -Table 2?

Author response 6: We have revised the numbering; the suggested changes have been made.

  1. I highly recommend a revision of the tex by native English language speaker since there are some parts that present several languages problems.

Author response 7: Thank you for your comment, we have taken into account the revision of the manuscript.

Round 2

Reviewer 1 Report

All concerns addressed in the revised manuscript.